# Mitigating Privacy Risk of Adversarial Examples with Counterfactual Explanations

## Abstract

Robustness and privacy are two fundamental security properties that machine learning models require. Without balance between robustness and privacy leads to robust models with high privacy risks. Obtaining machine learning models with high adversarial robustness and privacy performance remains an open problem. In order to enhance privacy performance of robust models, we employ counterfactual explanations as a method to mitigate privacy risks while concurrently maintaining robust model accuracy, reducing the privacy risk of the robust model to the level of random guessing and using counterfactual explanations to generate adversarial examples for the first time. We analyze the similarities and differences between adversarial examples and counterfactual explanations and utilize these properties to design the generation method. We conduct an in-depth analysis of the advantages offered by counterfactual explanations compared to traditional adversarial examples. Our study indicates that the correlation between robustness and privacy is strong and the ideal balance state of accuracy, robustness, and privacy is with 95% adversarial examples involved in model training.

## 1 Introduction

Adversarial attack is adding an imperceptible perturbation to the input image that leads a machine learning model to misclassify the perturbed input image with high confidence (Goodfellow et al., 2015). Adversarial examples are inputs that have been perturbed by attackers. To build adversarially robust machine learning models that can defend adversarial attacks, a variety of defense strategies have been put forth (Moosavi-Dezfooli et al., 2016; Carlini & Wagner, 2017; Madry et al., 2018). Adversarial training, a training scheme that involves supplementing the training datasets with adversarial examples, is one of the most popular and effective defense strategies (Goodfellow et al., 2015).

With the growing development and application of adversarial example generation techniques in real-world settings, the issue of protecting privacy in data-analytic training has gained significant attention. Severe privacy issues are raised by privacy attacks, especially the privacy risks on robust models (Fredrikson et al., 2015; Ganju et al., 2018; Salem et al., 2020). Compared to non-robust models trained on original examples, robust models trained on adversarial examples pose a greater risk to privacy (Song et al., 2019). Defenses against such attacks on robust models have proven largely ineffective, despite higher risks.

To enhance the non-robust model privacy performance, a variety of techniques are frequently employed (Huang et al., 2021; Liu et al., 2022). When privacy alone is taken into account, they function perfectly. However, these techniques weaken the ability of robust models to predict outcomes, making the models unable to function normally (Song et al., 2019). The privacy risk caused by adversarial examples remains an open problem.

The connection between adversarial robustness and privacy received significant interest. Recent discussions suggest that there may be a fundamental trade-off, as achieving both robustness and privacy within the same model appears unfeasible (Zhang et al., 2019; Hopkins et al., 2023). A primary factor contributing to privacy risks is the generalization gap between adversarial and original examples (Yeom et al., 2018). Privacy attackers can readily detect differences in distribution between these two types of examples, which increases privacy risks in robust models.

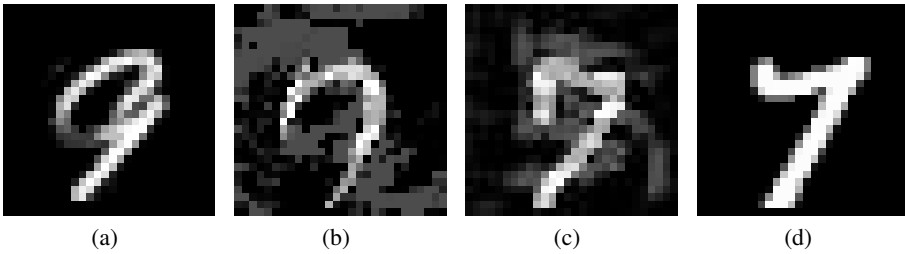

(a)             (b)             (c)             (d)

Figure 1: Adversarial examples generated by our method (a) are closer to the original sample (d) with better privacy performance. The perturbations of adversarial examples generated on grayscale image datasets by previous methods such as (b) and (c) are more obvious and perceptible. The perturbation is inconsistent with the definition of adversarial examples. (a) Counterfactual adversarial example (b) PGD adversarial example (c) AdvGAN adversarial example (d) MNIST dataset

Moreover, adversarial perturbations added on adversarial examples are imperceptible as its definition (Goodfellow et al., 2015). When the same adversarial example generation method is used in the grayscale datasets and RGB datasets, the perturbations on the grayscale samples are obvious and meaningless. As shown in Figure 1 (b) and (c), these are the samples of adversarial examples generated by two different approaches, and the perturbations are far from imperceptible as a main characteristic of adversarial examples.

In this work, we propose generating adversarial examples that are aligned with the original data distribution. We achieve this by using counterfactual explanations to create in-distribution adversarial examples. We analyze and leverage these shared characteristics to enhance the privacy performance of adversarial examples. Furthermore, the adversarial examples produced are more semantic and meaningful due to the explainable nature of counterfactual explanations. We make the contributions as follows:

- We apply counterfactual explanations to generate adversarial examples for the first time. We design the counterfactual adversarial examples with the analysis of similarities and differences between counterfactual explanations and adversarial examples.

- We mitigate the privacy leakage of robust models during the algorithm design process for the first time. It includes separating adversarial example generating and model training processes, finding sparse adversarial examples, and projecting the samples into latent space with the autoencoder.

- We generate semantic adversarial perturbations instead of meaningless noise. It provides an important opportunity to advance the understanding of the differences between counterfactual explanations and adversarial examples of the same original sample.

## 2   Counterfactual Explanations and Adversarial Examples

In this section, we analyze the similarities and differences between counterfactual explanations and adversarial examples. By utilizing both, we generate counterfactual adversarial examples aimed at reducing the privacy risks in robust models.

Counterfactual explanation is a technique used in explainable AI. Counterfactual explanations suggest what should be different in the input instance to change the outcome of an AI system (Guidotti, 2024). Recently, it has been suggested that there is a similarity and connection between counterfactual explanations and adversarial examples (Pawelczyk et al., 2022; Freiesleben, 2022).

Given an image-label pair $(x, y)$ and a classifier $f$, an adversarial example is a perturbed image $\tilde{x} = x + \delta$, which is mis-classified by the model, i.e., $f(\tilde{x}) \neq y$, which is the untargeted adversarial example, $f(\tilde{x}) = \tilde{y}$, for targeted adversarial example.

For the same pair $(x, y)$ and the classifier $f$, a counterfactual explanation is a perturbed image $x' = x + \delta'$, $f(x') \neq y$, use the generating process to explain why $x$ belong to class $y$. $f(x') \neq y$ to explain the reason why $x$ belong to class $y$ other than class $y'$. The definitions for both untargeted

and targeted adversarial examples share similarities. The commonalities between counterfactual explanations and adversarial examples are as follows:

- Both involve adding perturbations to the original sample.
- The perturbed samples are intended to alter the model's prediction.

The generated sample displays commonalities between adversarial examples and counterfactual explanations in the initial stage of generating. However, as the process proceeds, counterfactual explanations emphasize interpretability, focusing on when and why the predicted classification changes. By the end of the generating process, the counterfactual explanation resembles another class. In contrast, the perturbation of the adversarial example remains imperceptible to humans, in other words, seeking uninterpretability. The new features of the adversarial example are strong enough to change model prediction, but the adversarial example still looks similar to the original sample. While the new features of counterfactual explanation should be obvious and interpretable for humans, and its constraint of minimal perturbation can be negligible without considering computation efficiency.

Moreover, adversarial examples and counterfactual explanations have different constraints. For counterfactual explanations, interpretability is key, meaning the altered sample must be noticeably distinct from the original. In contrast, adversarial examples involve imperceptible perturbations. While both forms of perturbation shift the predicted class, the extent of the sample modification varies. The crucial distinction is that a counterfactual explanation is visibly different from its original, while an adversarial example closely resembles the original.

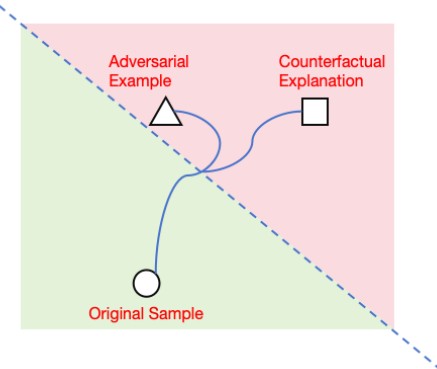

Figure 2: An illustration for the difference between the adversarial example and the counterfactual explanation generated from the same original sample

To clarify the relationship between counterfactual explanations and adversarial examples, we use a simple picture to demonstrate, as shown in Figure 2. Adversarial examples and counterfactual explanations both change the class of the sample, so they both cross the dicision boundary. At some point in the generation process, adversarial examples and counterfactual explanations may share the same tendency in direction to change key features for class alteration. But their constraints differ, so there are distance between the generated samples.

By considering the similarities and differences between counterfactual explanations and adversarial examples, we create counterfactual adversarial examples. This involves locating the nearest neighbor of the original sample from another class. We generate adversarial examples of the original by producing counterfactual explanations of the neighbor sample. During the counterfactual generation, the features of the generated sample shift from the neighbor's attributes to resemble the original. The result is an adversarial example with characteristics of the original sample, while still retaining some features from the neighbor class. This ensures that the adversarial example incorporates features from the neighbor sample, enough to alter the model's prediction rather than being identical to the original sample.

To ensure privacy protection, we need an appropriate counterfactual explanation method. There are different types of counterfactual explanation generation methods (Guidotti, 2024). However,

not all of them effectively reduce privacy risks. The privacy concerns associated with generative models, such as GAN and others, remain a prominent issue (Hitaj et al., 2017; Hayes et al., 2019). Consequently, counterfactual explanation methods based on generative models are excluded from consideration. In contrast, autoencoders offer improved privacy performance, making them the best choice for generating counterfactual explanations.

Autoencoders enhance data privacy by obfuscating sensitive information while retaining essential features for various applications. Several studies propose innovative approaches utilizing autoencoders to address privacy concerns in different domains (Ding et al., 2023; Jamshidi et al., 2024; Liu et al., 2023). Samples generated in the latent space exhibit features that are less perceptible to humans after decoded, aligning more closely with the definition of adversarial examples. Latent space enhances data privacy by inferring cluster locations and scales from connection numbers alone, eliminating the need for node-level data and protecting data privacy (Hajihassani et al., 2020). Additionally, leveraging a latent space navigation strategy can generate diverse synthetic samples while addressing privacy concerns, minimizing the risk of near-duplicates and supporting effective deep model training (Raja, 2024).

To keep privacy risks minimal, we select a method that prioritizes protecting the privacy of the original samples. To reduce the privacy risk of adversarial examples, the key is to decrease the generalization gap between the adversarial example dataset and the training dataset. We use a sparsity loss term in the counterfactual explanations to make the data distribution of adversarial examples more similar to the original dataset. This approach reduces the privacy risk cause by disparity between these two data distributions. Sparsity can enhance privacy in neural networks. Research has shown that introducing sparsity in neural networks can bolster their data privacy, ultimately leading to improved privacy without compromising task performance (Chen et al., 2020).

Another reason for increased privacy risks in adversarial examples is the privacy leakage of individual data. The adversarial example loss term is usually calculated on the same individual sample iteratively, and the prediction model is more likely to remember the sample in the process of adversarial example generating and model training. We use the counterfactual explanation method that calculates the prototype of every classification instead, and the loss term can guide the algorithm to generate the adversarial examples close to the average shape of the targeted classification, instead of the ones that easily cause privacy leakage of individual data.

## 3 COUNTERFACTUAL ADVERSARIAL EXAMPLES

### 3.1 COUNTERFACTUAL ADVERSARIAL EXAMPLES

For an original sample $x$ with ground truth $y$, we need to find an adversarial example $x_{adv} = x + \delta$ , with both $x_{adv}$ and $x \subseteq \mathbb{R}^D$ where $\mathbb{R}^D$ represents the $D$-dimensional feature space. To ensure the robustness of the model, adversarial examples are employed in the model training process, with an objective function as Equation 1 shows. It means that the prediction $f(x_{adv})$ is changed into another class with the minimal perturbation added to sample $x$.

$$\arg\max_i f_i(x_{adv}) \neq y_i \tag{1}$$

We changed the starting point of the computation process due to the differences between adversarial examples and counterfactual explanations. We identify the neighbor sample $x_{nb}$ in the training set that is closest to the sample $x$ but belongs to a different class, designating it as the neighbor sample. The autoencoder utilizes an encoder to project the sample $x$ from the feature space onto the latent space, resulting in $g(x)$. Through a decoder, $g(x)$ can be restored to the sample $x$ in the original feature space. $g(x)$ and $g(x_{nb})$ should be as close as possible to generate the adversarial examples that look similar to the original sample $x$.

$$\min ||g(x) - g(x_{nb})||_2 \tag{2}$$

By generating counterfactual explanation $cf(x_{nb})$ of the neighbor sample $x_{nb}$, it produces the adversarial example $x_{adv}$ for the original sample $x$. The counterfactual explanation $cf(x_{nb})$ have the characteristic of the original sample $x$ due to the aim of the counterfactual explanation generation

method, produce samples that are similar to the targeted sample. The whole process enhances sparsity of the adversarial example distribution, fostering greater similarity between the data distribution of the adversarial example dataset and the original training dataset. The approach aims to safeguard the data privacy of the original samples.

$$x_{adv} = cf(x_{nb}) \tag{3}$$

## 3.2 MITIGATING PRIVACY RISK OF ADVERSARIAL EXAMPLES

### 3.2.1 PRIVACY AND GENERALIZATION GAP

To mitigate data privacy risks of adversarial examples, it is essential to minimize the generalization error, which is the prediction accuracy difference between the training and the corresponding test datasets.

$$\min(\frac{\sum_{x \in D_{train}} \mathbb{I}(f(x) = y)}{|D_{train}|} - \frac{\sum_{x \in D_{test}} \mathbb{I}(f(x) = y)}{|D_{test}|}) \tag{4}$$

$\mathbb{I}(\cdot)$ is the indicator function, utilized for statistically counting the accurate quantity of data.

To decrease the generalization error of the test dataset and the training dataset, we decouple the adversarial examples generating process from the model training process. The original training dataset is used to generate adversarial but remains unseen to the prediction model, so the prediction model will not remember the original training dataset. Adversarial examples are employed for robust model training, while the original training dataset is retained for model evaluation. Both training and test datasets are new data for robust models by the approach, thereby enhancing the privacy of robust models.

After the process above, the key problem is that the model performance varies between the original training dataset and the adversarial example dataset. Specifically, the model accuracy on the original training dataset is lower than that on the adversarial example dataset. To ensure more accurate predictions on the original training dataset, it is crucial to minimize the prediction error of the model on both non-robust examples and adversarial examples, as shown in Equation 5 and Equation 6. Because the training and the test dataset are both unfamiliar when the model is applied for prediction, these two equations can be equivalent when the data privacy is well protected in the adversarial example generation process.

$$\min(\frac{\sum_{x \in D_{train}} \mathbb{I}(f(x_{adv}) = y)}{|D_{train}|} - \frac{\sum_{x \in D_{train}} \mathbb{I}(f(x) = y)}{|D_{train}|}) \tag{5}$$

$$\min(\frac{\sum_{x \in D_{train}} \mathbb{I}(f(x_{adv}) = y)}{|D_{train}|} - \frac{\sum_{x \in D_{test}} \mathbb{I}(f(x) = y)}{|D_{test}|}) \tag{6}$$

### 3.2.2 MITIGATING PRIVACY RISKS BY COUNTERFACTUAL EXPLANATIONS

We improved the counterfactual explanations method (Van Looveren & Klaise, 2021) to make the generated samples with more characteristics of adversarial examples and less privacy risk. The method produces samples towards the prototype of each class. Firstly, the counterfactual explanations generated by the method exhibit the imperceptible nature of adversarial examples. Secondly, these explanations are directed towards the prototype of the ground-truth $y$ which $x$ belongs to, rather than focusing solely on individual sample $x$. The samples produced by this approach have the average and collective characteristics of the whole class, without leaking the special and unnecessary characteristics of an individual sample. Moreover, the adversarial examples $x_{adv}$ generated by this approach encapsulate more meaningful features related to the ground-truth $y$ of sample $x$, rather than isolated characteristics of sample $x$ or perturbations as meaningless noise. This is the reason why the method can facilitate the acquisition of necessary features for adversarial examples and concurrently safeguard data privacy simultaneously.

$$L = L_{pred} + L_{AE} + L_{sparse} + L_{proto} \tag{7}$$

As shown in Equation 7, there are four main loss terms in counterfactual explanations, the loss term $L_{pred}$ for the correct model prediction, the loss term $L_{AE}$ for the transformation into target

classification, the loss term $L_{sparse}$ for the sparsity of the generated perturbation,and the loss term $L_{proto}$ for the prototype of the target label. More details about loss function in supplementary materials.

# 4 EXPERIMENT

## 4.1 EXPERIMENT SETUP

We utilize CNN as the machine learning model architecture. The CNN model structure used in the experimentcontains three dropout layers, as greater dropout layers often yield better generalization effects (Song et al., 2019). We ensure consistency in each experiment by employing the same model architecture for both adversarial example generation and training processes. The models are all fully trained but do not overfit with the training datasets to reduce the impact of the generalization gap as much as possible. We separate the adversarial examples generation and training process of all generation methods instead of put the two processes together as the original generation method to mitigate the impact of the machine learning model itself. To assess the privacy risk of these CNN models, we employ a membership inference attack (MIA). MIA attackers aim to determine whether given data belongs to a training set (Shokri et al., 2017). It is a technique widely used to evaluate how much privacy is compromised during the training process. All experiments are conducted under the black-box MIA setting. The attacker model can only achieve the prediction results of the training model. Our research aims to reduce the privacy risk of adversarial examples and explore the relationship between robustness and privacy, contrasting with white-box MIA studies focusing on the model's inherent data privacy leakage. We utilize a state-of-the-art attack method specifically designed for robust models (Song et al., 2019).

## 4.2 EXPERIMENT RESULTS

Table 1: Data point quantities of adversarial examples and original training dataset

| ADVERSARIAL EXAMPLE METHOD | QUANTITIES |
|---|---|
| **OUR METHOD** | **53539** |
| PGD | 55000 |
| ADVGAN | 11579 |
| ORIGINAL MNIST DATASET | 60000 |

We compare the number of adversarial examples generated by different methods. Table 1 illustrates the contrast between the quantities of adversarial examples generated by all experiment methods and the original MNIST training dataset. The MNIST dataset consists of 60,000 training samples. To ensure that the total number of data is roughly equal to our method, we generate 55,000 adversarial examples with PGD method, and our method generates a total of 53,539 adversarial examples. In contrast, the AdvGAN method produced only 11,579 adversarial examples.

Our counterfactual adversarial examples method reduced the MIA accuracy to 50.00%, extremely approaching the minimal random guessing probability. It is even lower than the MIA accuracy for non-robust models, which stood at 50.46%. In comparison with other robust models, our model maintained the lowest membership inference attack accuracy and achieved the highest training and testing accuracy in the meantime, as indicated in Table 2. The PGD method can produce the largest quantity of adversarial examples, but the accuracy and the privacy performance are not as good as other methods. The AdvGAN method achieved quite a good performance on accuracy and privacy risk with the least samples among all datasets, but the method based on the GAN model is more vulnerable to membership inference attack (Hayes et al., 2019). Therefore, if the quantity of the adversarial examples generated by the AdvGAN method is closer to the MNIST dataset, the privacy performance may be not as good as the other methods. The result indicates that our method has the best performance on accuracy and privacy among these adversarial example generation methods. It not only verifies that the counterfactual explanation method can reduce more privacy risks than previous methods, but also shows that the counterfactual explanation method is truly promising for improving adversarial example generation.

Table 2: Accuracy of training, test and membership inference attack with different adversarial example methods

| ADVERSARIAL EXAMPLE METHOD | TRAINING ACCURACY | TEST ACCURACY | MIA ACCURACY |
|---|---|---|---|
| **XAE** | **73.56%** | **75.27%** | **50.00%** |
| PGD | 71.95% | 70.80% | 50.79% |
| ADVGAN | 73.35% | 73.11% | 50.21% |

## 4.3 PRIVACY RISKS DIFFENRENCE BETWEEN COUNTERFACTUAL ADVERSARIAL EXAMPLES AND ORIGINAL DATASETS

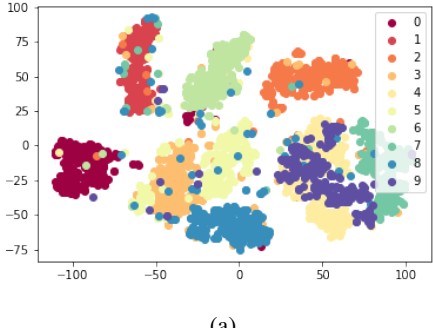
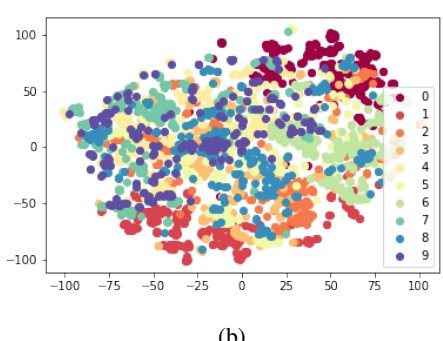

(a)  (b)

Figure 3: Comparison between original dataset and adversarial example datasets, (a) MNIST dataset (b) Counterfactual adversarial example dataset

We conduct further analysis to understand the performance differences between adversarial examples and original datasets on data privacy. They both are analyzed by the same t-SNE method with the same iteration (Van der Maaten & Hinton, 2008). The t-SNE method is a dimensionality reduction approach to analyzing samples. If the samples share more similarities than other samples, they are closer than the others in the result. Figure 3(a) shows the original similarity among all the labels on the MNIST dataset. All the samples are gathered by their ground truth because samples belonging to the same label are the most alike in contrast to the other labels. The samples in classes 3,5,8 look more alike in their shapes, as a result, these three labels are closer than other labels, same as classes 7,9,4. As shown in Figure 3(b), it is difficult for the t-SNE method to extinguish the disparities of the samples from different classes. One of the reasons is that our generating method starts at a data point that belongs to another class different from the original sample. This may explain why counterfactual adversarial examples have better privacy performance against MIA than the original datasets.

## 4.4 ANALYSIS OF PRIVACY RISKS ON DIFFERENT ADVERSARIAL EXAMPLE DATASETS

We conduct further analysis to understand the performance differences between different methods on data privacy. The t-SNE setting remains the same with the experiment in Section 4.3. It is evident that samples generated by the PGD method are more dispersed, showcasing clearer distinctions among various classes, as shown in Figure 4(a). The result can elucidate its higher membership inference attack accuracy. While the disparity is minor in some classes, significant disparities exist in others for the AdvGAN method as shown in Figure 4(b). Compared to other methods, the adversarial examples generated by our method are gathered together despite the different labels. As shown in Figure 4(c), it is difficult for the t-SNE method to extinguish the disparities of the samples from different classes. One of the reasons is that our method generates perturbations on latent space, other than the high-dimensional feature space. This may explain why our method produces adversarial examples with much lower privacy risk than other adversarial example generation methods.

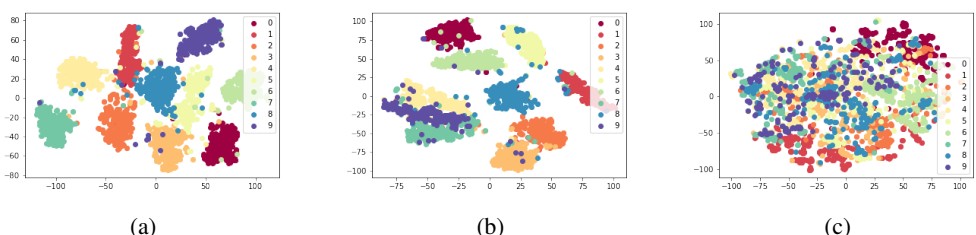

(a)             (b)             (c)

Figure 4: Data Distribution Comparison between different adversarial example methods, (a) PGD adversarial example dataset (b) AdvGAN adversarial example dataset (c) Counterfactual adversarial example dataset

## 4.5 THE BEST BALANCE AND TRADE-OFF BETWEEN ACCURACY, ROBUSTNESS AND PRIVACY

To find the best balance and trade-off between accuracy, robustness, and privacy, and analyze the correlation between robustness and privacy, we investigated the model's robustness and privacy by altering the proportion of adversarial examples in the training set. To decrease the effect of data point increment, we control the quantity of training data at 53539, the same as the data point amount in our adversarial example generating method. As privacy performance decreases, robustness notably increases, as illustrated in Table 3. With a larger the proportion of counterfactual adversarial examples, the model's privacy strengthens, while an increased the proportion of original samples correlates with higher model accuracy. However the correlation between model robustness and privacy is not strictly linear. The peak of training accuracy is with 10% counterfactual adversarial examples training the machine learning model, while the peak of test accuracy is with 20% counterfactual adversarial examples, rather than original samples. With 90% counterfactual adversarial examples training model, the privacy performance is at the same level as the original samples, but the robustness is highly reinforced with such a big proportion of adversarial examples. We believe that the model performance of accuracy, robustness and privacy achieved the desired balance with about 95% counterfactual adversarial examples to get much higher model performance on the accuracy, a reasonable decrease on robustness and a slight and acceptable increase in privacy risk.

Table 3: Accuracy of training, test and membership inference attack with different adversarial example methods

| ROBUST DATA RATIO | TRAINING ACCURACY | TEST ACCURACY | MIA ACCURACY |
|---|---|---|---|
| 0% | 99.42% | 98.80% | 50.46% |
| 10% | 99.77% | 98.83% | 52.05% |
| 20% | 99.66% | 98.95% | 51.63% |
| 30% | 99.5% | 98.68% | 51.64% |
| 40% | 99.40% | 98.69% | 51.38% |
| 50% | 99.18% | 98.54% | 51.42% |
| 60% | 98.92% | 98.44% | 51.18% |
| 70% | 98.66% | 98.17% | 50.98% |
| 80% | 98.01% | 97.66% | 50.76% |
| 90% | 96.65% | 96.45% | 50.38% |
| 95% | 94.91% | 94.92% | 50.07% |
| 99% | 87.58% | 88.46% | 50.01% |
| 100% | 73.56% | 75.27% | 50.00% |

## 5 CONCLUSION

In this paper, we aimed to create in-distribution adversarial examples by leveraging the similarities and differences between adversarial examples and counterfactual explanations to reduce the privacy risks of robust models. By utilizing counterfactual explanations from the nearest neighbor class, we generated adversarial examples with enhanced privacy. Our proposed counterfactual adversarial examples offer better privacy protection, more meaningful and semantic perturbations, and maintain an acceptable level of accuracy. Although we have improved the method for other kinds of training datasets and obtained certain effects, there are still deficiencies in the targeted adversarial examples, such as two classes of excessive distance. Constrained by the length of the paper, we will leave it as future work.

ACKNOWLEDGMENTS

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
