# OpenReview forum: "Mitigating Privacy Risk of Adversarial Examples with Counterfactual Explanations"
_ICLR.cc/2025/Conference — Submitted to ICLR 2025_

### Official Review · Reviewer_GcoQ · 2024-10-31

**Soundness:** 2
**Presentation:** 1
**Contribution:** 2
**Rating:** 3
**Confidence:** 4

**Summary:**

It is hard to implement a deep learning model that protects privacy and is robust at the same time. Specifically, adversarial examples can be used to achieve the purpose of robustness, but it is usually difficult to protect privacy on this basis. This paper analyzes the relationship between counterfactual explanations and adversarial examples, and designs the counterfactual adversarial example to mitigate the privacy leakage of robust models.

**Strengths:**

1. This article proposes the idea of reducing privacy risk, which is inspiring.

**Weaknesses:**

1. This article uses a lot of space in section 2 to describe the similarities and differences between counterfactual explanations and Adversarial examples, which have been elaborated in many articles [1,2]. Further analysis is needed on the relationship with privacy.
2. The formulas written in this article need to be further improved. Some mathematical symbols that appear for the first time (such as Eq. (7)) are not explained, and some formulas (such as Eq. (1) and (2)) do not correspond to the context, but appear abruptly in those positions.
3. The description of the method part of this article needs to be improved. Sections 3.1, 3.2, and 3.3 look more like three independent parts.
4. This article needs more experiments to reflect the effectiveness of the method. This article only made a CNN model on MNIST, and did not verify whether it is effective on larger datasets and more complex models.


[1]Pawelczyk, Martin, et al. "Exploring counterfactual explanations through the lens of adversarial examples: A theoretical and empirical analysis." International Conference on Artificial Intelligence and Statistics. PMLR, 2022.

[2]Jeanneret, Guillaume, Loïc Simon, and Frédéric Jurie. "Adversarial counterfactual visual explanations." Proceedings of the IEEE/CVF Conference on Computer Vision and Pattern Recognition. 2023.

**Questions:**

1. In the article, I saw that Autoencoders can reduce privacy risk, but GAN-generated ones cannot. According to the existing article, reducing privacy risk is related to the method of generating counterfactual examples, not the counterfactual examples themselves. So can adversarial examples generated by Autoencoders reduce privacy risk, rather than because of the nature of counterfactual examples? If it is because of the nature of counterfactual examples, can you explain why examples generated with counterfactual explanations can reduce privacy risk?
2. Can some mathematical formulas be improved to make it easier for readers to understand? Is $i$ in $f_i$ of Eq. (1) a class? Is Eq. (2) an optimization target?
3. Some mathematical formulas need more explanation. Are Eq. (4), (5), and (6) used in which part of the loss in generating counterfactual adversarial examples? What are the four losses in Eq. (7), and are they related to the previous formulas?

---

### Official Review · Reviewer_phZK · 2024-11-02

**Soundness:** 1
**Presentation:** 1
**Contribution:** 1
**Rating:** 3
**Confidence:** 4

**Summary:**

This paper proposes using counterfactual explanations to reduce privacy risks in robust machine learning models, aiming to balance robustness, accuracy, and privacy by generating adversarial examples through counterfactuals, ultimately achieving a privacy risk level comparable to random guessing.

**Strengths:**

- **Intersection between robustness & privacy is interesting**: The paper addresses a complex intersection between privacy and robustness in machine learning.

- **Training algorthim**: By attempting to incorporate counterfactual explanations into model training, the paper takes a new perspective to adversarial robustness and privacy alignment.

**Weaknesses:**

- **Relevance and Motivation**: The practical relevance of the proposed solution remains unclear. The motivation for focusing on counterfactual explanations over traditional methods for balancing robustness and privacy lacks justification, leaving doubts about its necessity.

- **Metric Selection for Privacy Evaluation**: The paper’s use of membership inference accuracy as a privacy metric is inadequate. A more suitable metric would be the true positive rate (TPR) at a low false positive rate (FPR), as this metric would allow an adversary to determine training set membership with higher confidence. Existing work, such as [7], highlights TPR @ low FPR as a more meaningful measure in privacy settings.

- **Missing Related Works and Literature Misrepresentation**: The paper lacks a clear related works section, making it difficult to contextualize its contributions within existing research. For example, it fails to adequately cite and compare itself to relevant counterfactual explanation methods (e.g., [1-4]), leaving its method selection ungrounded.
Some statements in the paper seem to misinterpret or misrepresent findings from prior research [8], diminishing the credibility of its claims. Specifically, the lack of distinction between counterfactual explanations and adversarial examples is problematic, as references like [6] demonstrate they can be equivalent, and [5] highlights the privacy risks that counterfactual explanations themselves can pose.

- **Lack of Clarity in Methodology**: Key details are missing regarding the generation of counterfactual explanations. The lack of specifics regarding the generation process and the unclear formalization of key equations (e.g., equations 2 and 4) make the methodology difficult to follow and replicate.
Definitions of critical terms, such as the exact differences between adversarial examples and counterfactual explanations, are either ambiguous or absent, leading to confusion about the novelty and benefits of the proposed approach.


Based on these weaknesses, the following **suggestions for improvement** could be considered:

- Clearly articulate the relevance of counterfactual explanations for privacy-robustness tradeoffs in machine learning and provide a detailed comparison with existing methods.
- Consider evaluating privacy using TPR at low FPR, as discussed in [7], and address potential limitations of using membership inference accuracy as the sole metric.
- Include a comprehensive related works section that connects to the literature on counterfactual explanations, especially work such as [1] and [5], to provide a more thorough foundation for the methodology.
- Improve clarity in the formalization of the method, specifically in explaining the setup for generating counterfactuals and addressing unclear notation in key equations.


----

**References**

[1] Carla: a python library to benchmark algorithmic recourse and counterfactual explanation algorithms, https://arxiv.org/abs/2108.00783

[2] Towards Realistic Individual Recourse and Actionable Explanations in Black-Box Decision Making Systems, https://arxiv.org/abs/1907.09615

[3] Learning model-agnostic counterfactual explanations for tabular data, https://arxiv.org/abs/1910.09398

[4] Getting a CLUE: A Method for Explaining Uncertainty Estimates, https://arxiv.org/abs/2006.06848

[5] On the Privacy Risks of Algorithmic Recourse, https://arxiv.org/abs/2211.05427

[6] Exploring Counterfactual Explanations Through the Lens of Adversarial Examples: A Theoretical and Empirical Analysis, https://proceedings.mlr.press/v151/pawelczyk22a.html

[7] Gaussian Membership Inference Privacy, https://proceedings.neurips.cc/paper_files/paper/2023/hash/e9df36b21ff4ee211a8b71ee8b7e9f57-Abstract-Conference.html

[8] Robustness Implies Privacy in Statistical Estimation. In Proceedings of the 55th Annual ACM Symposiumon Theory of Computing, pp. 497–506, 2023.

**Questions:**

See above.

---

### Official Review · Reviewer_XVyY · 2024-11-04

**Soundness:** 2
**Presentation:** 2
**Contribution:** 2
**Rating:** 3
**Confidence:** 3

**Summary:**

-   The paper tackles an important topic in the field of machine learning, specifically the tradeoff between privacy and performance in the context of adversarial examples.
-   The authors propose an approach to generating counterfactual explanations that are aligned with the original data distribution.

**Strengths:**

-   The paper tackles an important topic in the field of machine learning, specifically the tradeoff between privacy and performance in the context of adversarial examples.
-   The authors propose an approach to generating counterfactual explanations that are aligned with the original data distribution.

**Weaknesses:**

-   The paper lacks clarity and readability, with technical terms and acronyms (e.g. CNN) used without definition.
-   The comparison of different baselines is not well-explained, and the relevance of these baselines to the context is not clear.
-   The paper only uses the MNIST dataset and does not define the model architecture, which limits the generalizability of the results.
-   There are major spelling errors throughout the paper (for eg. title of Section 4.3 ), which detracts from its overall quality.
-   The comparison with state-of-the-art membership inference attacks (MIA) and variants is not discussed, which could help evaluate the potency of this method.

**Questions:**

-   Can the authors provide more clarity on the technical terms and acronyms used throughout the paper?
-  This paper will benefit from comparison with other state of the art generative methods to improve privacy , such as [1], [2]
-   Can the authors provide more information on the threat model for the membership inference attacks (MIA) used in the paper, and how they relate to the context of adversarial examples?

[1] Jia, Jinyuan, et al. "Memguard: Defending against black-box membership inference attacks via adversarial examples." Proceedings of the 2019 ACM SIGSAC conference on computer and communications security. 2019.

[2] Hu, Hailong, and Jun Pang. "Loss and Likelihood Based Membership Inference of Diffusion Models." International Conference on Information Security. Cham: Springer Nature Switzerland, 2023.

---

### Meta-Review · Area_Chair_riTB · 2024-12-21

**Metareview:**

The paper proposes to study privacy robustness tradeoffs by analyzing counterfactual examples and adversarial examples. By generating adversarial examples using counterfactual frameworks and imposing robustness, authors achieve better robustness-privacy tradeoffs. Reviewers have noted weak empirical evaluation due to insufficient datasets and the choice of metrics, lack of clarity in the writing and explaining methodology, and insufficent literature review. An author response was not provided. In going with reviewer consensus, I recommend a reject.

**Additional Comments On Reviewer Discussion:**

No additional discussions during discussions.

---

### Decision · Program_Chairs · 2025-01-22

Reject